# Fetal Exposure to Air Pollution in Late Pregnancy Significantly Increases ADHD-Risk Behavior in Early Childhood

**DOI:** 10.3390/ijerph191710482

**Published:** 2022-08-23

**Authors:** Binquan Liu, Xinyu Fang, Esben Strodl, Guanhao He, Zengliang Ruan, Ximeng Wang, Li Liu, Weiqing Chen

**Affiliations:** 1Nanjing Institute of Geography and Limnology, Chinese Academy of Sciences, Nanjing 210006, China; 2Department of Epidemiology and Biostatistics, School of Public Health, Anhui Medical University, Hefei 230032, China; 3Department of Public Health and Health Administration, Clincial College of Anhui Medical University, Hefei 230031, China; 4Anhui Province Laboratory of Inflammation and Immune Mediated Disease, Hefei 230032, China; 5School of Psychology and Counselling, Queensland University of Technology, Brisbane, QLD 4059, Australia; 6Department of Biostatistics and Epidemiology, School of Public Health, Sun Yat-sen University, Guangzhou 510080, China; 7Department of Information Management, Xinhua College of Sun Yat-sen University, Guangzhou 510080, China

**Keywords:** fetal exposure, air pollution, ADHD-like behavior

## Abstract

Background: Air pollution nowadays has seriously threatened the health of the Chinese population, especially in the vulnerable groups of fetuses, infants and toddlers. In particular, the effects of air pollution on children’s neurobehavioral development have attracted widespread attention. Moreover, the early detection of a sensitive period is very important for the precise intervention of the disease. However, such studies focusing on hyperactive behaviors and susceptible window identification are currently lacking in China. Objectives: The study aims to explore the correlation between air pollution exposure and hyperactive behaviors during the early life stage and attempt to identify whether a susceptible exposure window exists that is crucial for further precise intervention. Methods: Based on the Longhua Child Cohort Study, we collected the basic information and hyperactivity index of 26,052 children using a questionnaire conducted from 2015 to 2017, and the Conners’ Parent Rating Scale-revised (CPRS-48) was used to assess hyperactive behaviors. Moreover, the data of air pollution concentration (PM_10_, PM_2.5_, NO_2_, CO, O_3_ and SO_2_) were collected from the monitoring station between 2011 to 2017, and a land-use random forest model was used to evaluate the exposure level of each subject. Furthermore, Distributed lag non-linear models (DLNMs) were applied for statistic analysis. Results: The risk of child hyperactivity was found to be positively associated with early life exposure to PM_10_, PM_2.5_ and NO_2_. In particular, for an increase of per 10 µg/m^3^ in PM_10_, PM_2.5_ and NO_2_ exposure concentration during early life, the risk of child hyperactivity increased significantly during the seventh month of pregnancy to the fourth month after birth, with the strongest association in the ninth month of pregnancy (PM_10_: OR = 1.043, 95% CI: 1.016–1.071; PM_2.5_: OR = 1.062, 95% CI: 1.024–1.102; NO_2_: OR = 1.043, 95% CI: 1.016–1.071). However, no significant associations among early life exposure to CO, O_3_ and SO_2_ and child hyperactive behaviors were observed. Conclusions: Early life exposure to PM_10_, PM_2.5_ and NO_2_ is associated with an increased risk of child ADHD-like behaviors at the age around 3 years, and the late-prenatal and early postnatal periods might be the susceptible exposure windows.

## 1. Introduction

Nowadays, air pollution is nominated by the WHO as one of the most significant health threats. An expanding body of evidence suggests a link between long-term air pollution exposure and adverse health effects and increased mortality [1,2,3]. Recently, exposure to air pollutants has been proposed as a potential risk factor for neurodevelopmental problems [4,5,6]. 

Inhaled air pollutants may deposit in the alveolar region of the lungs, where they cause oxidative stress and systematic inflammation [7,8]. They can also translocate into the systematic circulation, reaching the brain where microglia can be activated via the blood–brain barrier or the olfactory epithelium, and neurotoxicity and neuronal damage can be initiated when this activation occurs chronically. 

Because the brain is undergoing extensive cellular differentiation and growth during prenatal and early postnatal stages, fetal life and early childhood are considered periods of particular vulnerability to air pollutants that induce neurotoxic insults [9]. Experimental studies on animals suggest that air pollution exposure during early life periods causes neurotoxicity [10,11]. Observational data on children have also shown that the brain exposed to air pollution can respond quickly to the hazardous environment, laying the foundation for changes in the structure and volume of organs in the brain [12], such as the hippocampus, which is essential for the development of behavior and cognitive function [13].

Attention-Deficit/Hyperactivity Disorder (ADHD) is a prevalent behavioral disorder in children, characterized by symptoms of hyperactivity, inattention and impulsiveness [14]. It is common with childhood onset, and for the appropriately 60–70% symptoms, it persists into adolescence and adulthood, affecting individuals across their life span [15]. Children with ADHD symptoms are more likely to suffer from adverse life events. Studies [16,17,18,19,20] have implicated that ADHD is associated with decreased academic performance, more limited employment prospects, increased risk of criminality, drug abuse and relationship difficulties in later life, which cause severe social and economic burdens. Over the past three decades, the prevalence of ADHD among children in China has risen from 5.5% to 6.7% [21]. These rates of prevalence make the disorder a vital health problem in children in China. However, the etiology of ADHD remains enigmatic. While it is believed that ADHD is highly heritable [22], environmental factors that act as a trigger also play a vital role in the disorder [23]. Experts have recognized that intrauterine factors, parent–child interactions, cognitive stimulation and family socioeconomic status are all crucial to neurodevelopment [14]. However, air pollution brain effects rooted in fetal life and childhood are not generally acknowledged, especially in China.

To date, only some developed countries have conducted studies to investigate whether air pollution exposure during early life is associated with ADHD or symptoms of this disorder, but the results were conflicting [24,25,26,27,28,29,30,31,32,33]. Several studies [27,28,29,30,31,32,33,34] have provided some indications that air pollution exposure prenatally or during the early life stage—especially exposure to suspended PM, NO_2_, carbon monoxide (CO) and sulfur dioxide (SO_2_)—were associated with ADHD symptoms in children, while a null association has been found in other research [24,25,26]. 

Additionally, further identifying critical exposure windows for air pollutants and ADHD symptoms is of great importance for timely public prevention and intervention efforts to reduce the risk of ADHD. Although it is believed that the prenatal and early postnatal periods are the most vulnerable periods for neurodevelopment, previous studies [24,25,26,27,28,29,30,31,32,33,34] offered little evidence of the exposure window that reflects the most vulnerable period. The air pollution exposure data in these studies are limited by once-annual measurements, short-term measurements, or extrapolated data relying on monitoring stations intentionally located in highly polluted areas. Additionally, the correlations between different exposure times are usually ignored in data analysis. All of these reasons may lead to the limited recognition and misjudgment of the critical window identification. Fortunately, with the development of statistical methods, primarily distributed lag models (DLMs) [35], we are now able to determine more precise susceptibility windows on a fine scale. Additionally, we collected continuous information about daily concentrations of air pollutants from pre-pregnancy to post-birth ranging from 0–4 years of age in this study.

However, the links between air pollution exposure and ADHD symptoms might differ in other locations because of exposure characteristics (e.g., spatial differences in exposure measurement error) and/or features of the study population (e.g., race or ethnicity and education level). To clarify the potential role of air pollutants in China, we examine whether exposure to higher concentrations of air pollutants during early life is associated with the development of ADHD, based on the Shenzhen Longhua Child Cohort Study. In addition, we further identify the precise periods of exposure that may represent a sensitivity exposure window.

## 2. Methods 

### 2.1. Study Population and Design 

The current research was based on Longhua Child Cohort Study (LCCS), a large-scale population-based ongoing prospective cohort study located in Shenzhen City, China. LCCS was initiated in October 2014, aimed at exploring and evaluating the impact of family and school environments in early life on children’s neurodevelopment, particularly hyperactive behaviors, autistic-like behaviors and conduct problems. All the children, excluding those with mental or physical disorders, were recruited to the LCCS when they first entered the 171 kindergartens in Longhua District. Detailed information about LCCS has been mentioned in our previous studies [36,37]. This study involved the analysis of the baseline survey of LCCS during 2015–2017. More details on the inclusion and exclusion criteria of the study population can be observed in Figure 1. The study was approved by the Ethics Committee of the School of Public Health of Sun Yat-Sen University (2015–2016), and written informed consent was signed by all the children’s primary caregivers.

### 2.2. Questionnaire Survey

A well-trained healthcare assistant interviewed the parents by using a self-administered structured questionnaire that included socio-demographic characteristics of the children and parents (e.g., age, sex, marital status, education level and birth address), information before and during pregnancy (such as consumption of multivitamins, folic acid, calcium use, alcohol and engagement in active smoking), household air pollution condition (including exposure to fumes from cooking, environmental tobacco smoke (ETS), home renovation, mosquito coils and the burning of incense indoors), reproductive history (including preterm birth, low birth weight and delivery way) and postnatal condition (including feeding pattern, average daily sleep duration and frequencies of parent–child interactive activities). Further details about average daily sleep duration and parent–child interactive activities can be observed in our previous study [38].

### 2.3. Air Pollution Exposure Assessment

To derive residential exposure to air pollution (SO_2_, NO_2_, CO, O_3_, PM_2.5_, PM_10_), we used a land-use random forest (LURF) model developed by Shin Araki and colleagues [39]. Data on six air pollutant concentrations in Pearl River Delta were collected from 57 monitoring stations in the China National Environmental Monitoring Station (http://www.cnemc.cn/, accessed on 13 May 2020) and calculated based on the Ambient Air Quality Standards (GB 3095-2012) (China M E P, 2012). China began to perform the long-term monitoring of PM_2.5_, CO and O_3_ concentrations in 2013; the concentrations during 2011–2012 were backtracked by the deep learning multi-output LSTM (DM-LSTM) model. The geospatial inputs for the LURF model included (a) land-use data obtained by interpreting Landsat 8 images; (b) traffic data obtained from OpenStreetMap (https://www.openstreetmap.org, accessed on 13 May 2020); (c) population density data extracted from the Statistical Yearbook of Guangdong Province; (d) pollutant emission sources identified from the Information disclosure Platform of Guangdong Provincial Department of Ecology and Environment (http://gdee.gd.gov.cn/jcsj, accessed on 13 May 2020) and (e) elevation data obtained from a 250 m DEM from the Data Center for Resources and Environmental Sciences, Chinese Academy of Sciences (RESDC) (https://www.resdc.cn, accessed on 13 May 2020). 

We overlaid the spatial distribution of the birth address and the monthly spatial distribution grid layer of the atmospheric pollutant concentration in the study area to obtain the monthly time series data set of the atmospheric pollutant concentration of each subject. Detailed information for the LURF process and methods can be observed in the Appendix A.

### 2.4. Measurement of ADHD-like Behaviors

The Conners’ Hyperactive Index (HI), which is a subscale of the Conners’ Parent Rating Scale-revised (CPRS-48), was developed for the identification of hyperkinetic children in this study. CPRS-48 is a revised scale of the first version of the Conners’ Parent Rating Scale and consists of 48 items in a six-factor structure (conduct problems, learning problems, psychosomatic, impulsive–hyperactive, anxiety and hyperactivity) [40]. To date, it is widely validated and utilized across a range of international studies to assess ADHD-like behaviors in children aged between 3–17 years [41,42]. Moreover, Su et al. established norms for the CPRS in Chinese urban children in 2001 and demonstrated to have good reliability and validity [41] The HI subscale includes 10 items, which are: excitable, impulsive; cries easily or often; restless in the “squirmy” sense; restless, always up and on the go; destructive; fails to finish things; distractibility or attention span a problem; mood changes quickly and drastically; easily frustrated in efforts and disturbs other children [43,44,45]. Moreover, it used a 4-point scale from 0 (never) to 3 (often), and the final score was a mean score with a higher score indicating a high level of hyperactivity. Considering that age and sex were found and considered as a significant determinant of the scores on the HI, and some cultural differences have been found in other psychopathology screening measures between different countries, we used the 90th percentile HI score for the child’s age and gender as the cut-off instead of the same cut-off value of 1.5 for all children [41,42,46,47,48]. We strongly believe that this is a much more valid approach and significantly reduces misclassification errors. The Cronbach’s α coefficient for the HI was 0.83 in our study.

### 2.5. Statistical Analyses 

Continuous variables were described in terms of the mean (SD) or median (quartile), while categorical and ordinal variables were described as absolute frequencies and proportions. *T*-tests or chi-squared tests were used where appropriate for comparisons between groups. Since some HI scores contained “0” values, we added the value of “1” to the HI score and performed a natural logarithmic transformation to reduce the skewness of the data.

To simultaneously explore the time-varying effects and exposure-response of outdoor air pollution exposure throughout early life (first three years of life) on child hyperactivity at around 3 years old and identify respective vulnerable windows, we fitted distributed lag non-linear models (DLNMs) that can describe complex exposure-lag response by creating a cross-basis function [49]. In this study, a linear function was used to model the exposure–response relationship and a natural cubic spline function was utilized to estimate the lag-response association between six atmospheric pollutants and hyperactivity as continuous and categorical variables, respectively. A natural cubic spline function was selected due to its flexibility as well as the requirement for parsimony [49], and we assumed that the correlation between atmospheric pollutants and hyperactivity was smooth across months for the lag-response associations. The optimal degree of freedom of the lag structure was selected according to the minimum Akaike Information Criterion. Considering that the early life of children aged 3 years includes 10 months of pregnancy, 12 months of 0–1 year old and 24 months of 1–3 years old, we set 46 months as the maximum lag period [49]. Cumulative effects were further calculated to understand how serious it would be if one child suffered from consistent exposure to air pollutants in the first three years of life [49]. 

The covariates included child’s sex and age, maternal and paternal education, maternal and paternal age at the time of birth, family income, marital status, birth weight, parity, delivery way, feeding pattern, maternal passive smoking during pregnancy, gestation alcohol consumption, multivitamins, folic acid or calcium supplementation during pregnancy, gestational diseases (including gestational diabetes mellitus, pre-eclampsia/eclampsia and gestational hypertension), average daily sleep duration for children and frequencies of parent–child interactive activities at 0–1 and 1–3 years old and the number of persons in the house. In addition, we adjusted household air pollution conditions, including prenatal exposure to fumes from cooking, environmental tobacco smoke (ETS), home renovation, mosquito coils, the burning of incense indoors and exposure to ETS at 0–1 and 1–3 years old. Additionally, the monthly mean ambient temperature during the first three years of life was adjusted by using a natural cubic spline.

### 2.6. Sensitivity Analysis

The degree of freedom of lag distribution for six ambient air pollutants’ exposure was changed from 5–7 in the single-pollutant models to test the robustness. Pearson’s correlation was used to estimate the collaborations among six atmospheric pollutants and temperature for all participants. Additionally, the two-pollutant model was applied according to Pearson’s correlation test results to test the robustness of the single-pollutant model effect. Leave-one-out cross-validation (LOOCV) was conducted to evaluate the explained variance of the LURF model. Each of the measurements sites was sequentially left out by keeping the predictors and coefficients of the included predictors unchanged.

The statistical analyses were performed using R version 3.4.0, Statistics Analysis System (SAS, version 9.3, SAS Institute Inc., Cary, NC, USA), ARCGIS10.2 and MATLAB 7.0. All of the *p*-values were two-sided. Type-I errors were set at 0.05.

## 3. Results 

### 3.1. General Characteristics of Participants

A total of 26,052 participants were included in the final analyses, with an average age of 3.51 (*SD* = 0.29) years, and tended to be boys (54.3%) rather than girls (45.7%). Mean maternal age at childbirth was 28.01 (*SD* = 3.97) years (Table 1). The Conners’ HI scores ranged from 0 to 3, with a mean HI score of 0.44 (*SD* = 0.38) points. According to the standard cut-off value, 11.4% (2975/26,052) of the participants were over the threshold of a high probability of hyperactive behaviors. The mean values of monthly PM_10_, PM_2.5_, NO_2_, O_3_, SO_2_, CO and temperature during the entire study period were 54.57 μg/m^3^, 36.16 μg/m^3^, 34.63 μg/m^3^, 93.95 μg/m^3^, 8.98 μg/m^3^, 0.84 mg/m^3^ and 22.93 °C, respectively (Table 2). 

### 3.2. Effects of Air Pollution Exposure on Child Hyperactivity

Overall, exposure to PM_10_, PM_2.5_ and NO_2_ was associated with higher odds of having hyperactive behaviors at around 3 years old, while these results were not repeated in early life exposure to CO, O_3_ and SO_2_ (Table 3). The single-pollutant models in Figure 2 and Figure 3 illustrate that the risk of child hyperactivity at the age of 3 years was positively associated with PM_10_ and PM_2.5_ exposure during the 7th month of pregnancy to the 4th month after birth, which might be the susceptible exposure window, and the strongest association was observed during the 9th month of pregnancy (PM_10_ increased per 10 µg/m^3^, OR = 1.043, 95% CI: 1.016–1.071; β = 0.005, 95% CI: 0002–0.006. PM2.5 increased per 10 µg/m^3^, OR = 1.062, 95% CI: 1.024–1.102; β = 0.006, 95% CI: 0.002–0.008). Analogously, we observed a positive effect of NO_2_ exposure on child hyperactivity from the 7th month of pregnancy to the 3rd month after birth, with the strongest correlation during the 9th month of pregnancy (NO_2_ increased per 10 µg/m^3^, OR = 1.043, 95% CI: 1.016–1.071; β = 0.005, 95% CI: 0002–0.006. PM_2.5_ increased per 10 µg/m^3^, OR = 1.062, 95% CI: 1.024–1.102; β = 0.006, 95% CI: 0.002–0.008). However, no associations were shown between monthly CO, O_3_ and SO_2_ exposure and child hyperactive behaviors (Figure 4 and Figure 5). Similar results were found in the models in which the exposure to six ambient air pollutants was divided by quartiles (Figure 2, Figure 3, Figure 4 and Figure 5). Moreover, for a per 10 µg/m^3^ increase in PM_10_, PM_2.5_ and NO_2_ cumulative exposure during the sensitive exposure period, the risk of having hyperactive behaviors increased significantly, with the odds ratio of 1.315 (1.145–1.511), 1.335 (1.188–1.500) and 1.385 (1.126–1.703), respectively. Meanwhile, the ln (HI score + 1) increased by 0.031 (0.021–0.039), 0.033 (0.019–0.048) and 0.039 (0.023–0.056) points for each 10 µg/m^3^ increase in PM_10_, PM_2.5_ and NO_2_, respectively. 

### 3.3. Sensitivity Analyses 

We performed several sensitivity analyses to confirm our findings. Firstly, the two-pollutant models were constructed according to the correlation coefficient (r < 0.7) and we found similar results to those obtained for the single-pollutant models (Appendix A). Secondly, the *dfs* of the lag distribution of PM_10_, PM_2.5_ and NO_2_ exposure was changed in the DLNM models, and we found that our results did not significantly change (Appendix A). 

## 4. Discussion

In the present study, we assessed the relationship between early life ambient air pollution exposure, including PM_10_, PM_2.5_, NO_2_, CO, O_3_ and SO_2_, and hyperactive behaviors among 26,052 children of around 3 years old based on the LCCS. With a precise resolution and full adjustment, including household air pollution indicators, we found a more pronounced increase in the risk of hyperactivity in the child with higher PM_10_, PM_2.5_ and NO_2_ exposure, and the seventh month of pregnancy to the fourth month after birth might be the susceptible exposure window. 

Previous epidemiological studies assessing exposure to ambient air pollution during early life and child ADHD-like behaviors are scarce and present contradictory results. Siddique et al. conducted a cross-sectional study in 2011 that found a four-fold increase in the prevalence of ADHD among children residing in Delhi compared to children residing in less polluted rural areas of the country [31]. Newman et al. indicated that exposure to carbon attributed to traffic (ECAT) during infancy increased ADHD behavioral scores at seven years of age in Cincinnati City with a limit to children whose mothers had more than a high school education [30]. Yorifuji et al. in Japan analyzed data from singleton births with linked PM, NO_2_ and SO_2_ pollutant exposure during the nine months before birth and revealed a positive effect on aggressive behavior at the age eight years [32]. Min et al. examined the relationship between cumulative exposure to air pollutants (PM_10_ and NO_2_) from birth to diagnosis and childhood ADHD, and a positive correlation was uncovered [28]. Another study condcuted in Catalonia (Spain) with objective measures to the concentrations of outdoor and indoor school air pollutants, including black carbon (BC), ultrafine particle numbers (UFP; 10–700 nm) and PM_2.5_, demonstrated that cognitive development and attentiveness were reduced in children who were exposed to higher levels of air pollutants [50]. However, a survey based on the Child and Adolescent Twin Study in Sweden (CATSS) estimated residence time-weighted NOx and PM_10_ concentrations related to road traffic emissions during pregnancy in the first year, and the ninth year of life provided no support for a relationship between exposure to NOx or PM_10_ and ADHD in children [26]. Another collaborative study of eight European population-based birth/child cohorts also found that there was no evidence for an increase in the risk of ADHD symptoms with increasing prenatal PM and NO_2_ levels in children aged 3–10 years [25]. The substantial inconsistency and heterogeneity among the previously mentioned research are probably owing to the differences in geographic factors, participants’ size, data source, the accuracy of the exposure assessment as well as the air pollution situation in different countries. More specifically, some possible reasons might account for the discrepancies among the findings from Sweden, Japan, other European countries and our results: (1) relatively low levels of air pollution may contribute to the absence of a correlation and make it difficult to compare. The local concentrations of PM_10_ during the study period in Stockholm was only 3.9 µg/m^3^ [26] and the long-range PM_10_ in this region of Sweden has a yearly mean level of around 10 µg/m^3^ [51]. The air pollution levels merely ranged from 8.4 µg/m^3^ (Swedish) to 22.4 µg/m^3^ (Italian) for PM_2.5_ in the collaborative cohort study of European countries [25]. The levels of SO_2_, O_3_ and CO in the present research during the entire study period were, respectively, 8.98 µg/m^3^, 93.95 µg/m^3^ and 0.84 mg/m^3^, which is far below the National Environmental Quality Standard (60 µg/m^3^, 160 µg/m^3^ and 4 mg/m^3^, respectively) and lower than the average concentration of SO_2_ (13.43 µg/m^3^) in the study of Japan [32]. The data presented above can partly explain why no significant associations were observed among early life exposure to SO_2_, O_3_ and CO and child hyperactivity at the age of 3 years in this study. (2) Much more complex and larger proportions of possibly toxic components in air pollutants (PM) were found at a special increment concentration in the Pearl River Delta than those in other regions [52]. (3) The associations that exist between the socioeconomic status of the participants and the risk of behavioral problems may affect the results. As developed nations, residents in European countries are generally well-educated and have a high income, which may play a protective role in the onset of ADHD [53,54]. Moreover, a maternal, unhealthy lifestyle (including smoking and drinking) correlates with socio-economic status and may have contributed to this association [55]. 

The identification of susceptible exposure windows is crucial for understanding the causal mechanisms leading to symptom emergence and ADHD occurrence, as well as the timing of interventive measures implemented to reduce the risk and social burden of ADHD. Several studies [32,55], to date, have explored the sensitive windows for air pollution exposure during pregnancy associated with neurodevelopment delay. However, the approach those studies used to assess the effect of air pollution exposure was to regress the outcome on each trimester-average exposure separately in three regression models. Unfortunately, biased estimations might have occurred owing to the fact that exposure in different trimesters is always a closed correlated pattern and the effects of exposure of the given trimester lack the adjustment for exposure in other trimesters. Thus, we applied DLNM to identify the sensitive exposure period of air pollution during early life associated with hyperactive behaviors at the age of 3 years. The DLNM was proposed by Wilson et al. who compared different strategies to identify susceptible exposure windows, and finally pointed out that DLM can produce unbiased estimates and adds flexibility to identify windows [35]. Our study revealed that the seventh month of pregnancy to the fourth month after birth might be the susceptible exposure window, which can present direct evidence for precise interventions for hyperactive behavior.

Mechanisms for air pollution exposure on ADHD or behavioral problems are poorly understood, to date, but several hypotheses have been presented by researchers that seem to be suggest several concepts (1) Systematic inflammation. Air pollutants can trigger pro-inflammatory signals stemming from peripheral tissues/organs (lung and liver), giving rise to a release of circulating cytokines (such as TNF α and IL-1β) that transfers inflammation to the brain via olfactory, respiratory and blood–brain barriers [56,57]. Moreover, the systematic inflammatory caused by air pollution may be conducive to the deterioration of olfactory, respiratory and blood–brain barriers, and further enhance access to the brain and increase neuropathology [58]. (2) Particle effects and absorbed compounds. It is becoming increasingly accepted that ultrafine and fine particles (<100 nm) are the most notorious air pollutants [59]. Most reasons for this result from the properties ascribed to the physical and chemical composition of particulate matter, as well as its capacity to simulate immune immunity in the brain [60]. The nasal olfactory pathway is deemed to be the key threshold of access, where the inhalation of particulate matter has been observed to enter human frontal lobe and trigeminal ganglia capillaries [58]. In addition to the traits conferred by the physical and chemical constitution of the particulate matter, absorbed compounds present on particulate matter are mostly neurotoxic, which play a crucial role in neuropathology [61]. (3) Oxidative stress reaction. Air pollutants, especially ozones, are associated with brain effects. Acute or chronic exposure to ozones can induce oxidative stress and cytokine production in the brain, thus leading to brain lipid peroxidation, dopaminergic neuron death, neuronal morphological damage and motor deficits [62,63,64].

Concerning the susceptible exposure windows, there are several plausible mechanistic pathways for late-prenatal and early postnatal air pollution that contribute to the development of childhood hyperactivity. On one hand, the average flow velocity and blood flow of the umbilical artery increase significantly during the later trimester of pregnancy to fulfill the increasing demand for oxygen and nutrients in the fetus, which also increase the intensity of fetal exposure to other exogenous factors, such as air pollutants [65]. On the other hand, the formation of synapses and neurotransmitter both occur during the later trimester of pregnancy [43,66,67]. All air pollutants, regardless of matter inhaled during gestation or directly inhaled by the infant postnatally, can induce systematic inflammation and then alter microglia development, which plays a mediating role in synaptic pruning, thus resulting in altered neuronal connectivity and the disruption of the normal development of the brain. Further research is needed to identify more clearly and comprehensively the potential mechanisms. 

The results of the present study should be interpreted in the context of its limitations. First of all, all participants recruited in our study were limited to Shenzhen City, which may not represent the entire population and introduced selection bias. Secondly, the data of confounding variables in this study were retrospectively collected by a structured questionnaire, which might result in an information bias. Thirdly, air pollution exposure assessment was based on residential addresses instead of personal sampling, and daily behavior patterns during and after maternal pregnancy were not counted, thus leading to exposure misclassification to some degree. Fourthly, household pollution exposure was not measured directly in our study, as we aimed to examine the effects of individual levels of air pollution exposure on hyperactive behaviors; bias derived from household air pollution might have been considered. Additionally, although we adjusted for various potential confounding factors in our analyses, there was still the potential for residual confounding factors, such as a family history of ADHD. Therefore, further studies that include a larger study size, optimal research design and more accurate methods used to characterize air pollution level and toxicity are warranted to better understand the association of hyperactivity with ambient air pollutants.

## 5. Conclusions

We found that early life exposure to PM_10_, PM_2.5_ and NO_2_ was associated with an increased risk of child hyperactive e behaviors at the age of three years, and the late-prenatal and early postnatal periods might be susceptible exposure windows. If our findings are confirmed, the public health implications of these results would be of great importance by considering the optimal timing for the implementation of interventive measures and would further reduce substantial individual, family as well as social burdens accompanying hyperactive behaviors. 

## Figures and Tables

**Figure 1 ijerph-19-10482-f001:**
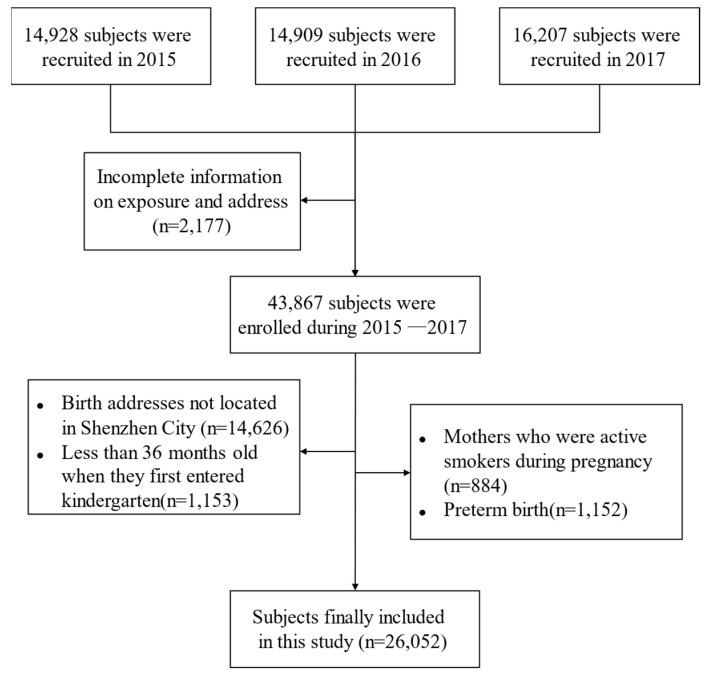
Flowchart of the study population inclusion.

**Figure 2 ijerph-19-10482-f002:**
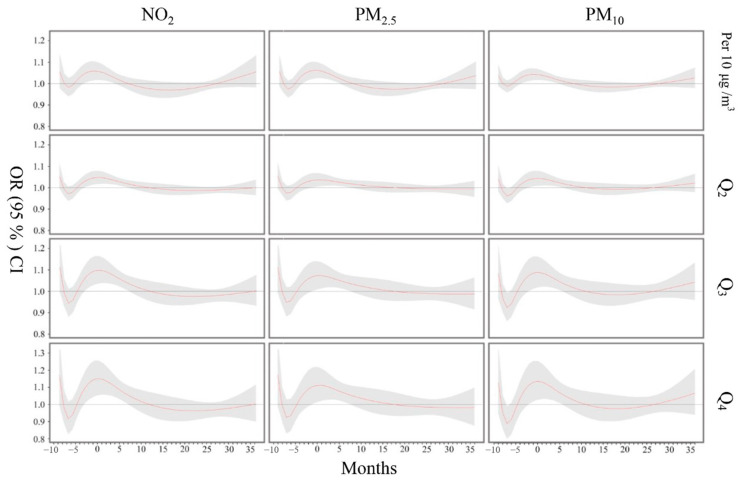
Odds ratio of child hyperactivity with monthly PM_2.5_, PM_10_ and NO_2_ exposure from fetal period to the first 3 years of life. Note: ORs (95% CI) in the three uppermost charts indicate the risks of PM_2.5_, PM_10_ and NO_2_ for each 10 μg/m^3^ increment in monthly PM_2.5_, PM_10_ and NO_2_ concentrations during pregnancy. ORs (95% CI) in the other charts indicate the risks of PM_2.5_, PM_10_ and NO_2_ for the second (Q2), third (Q3) and fourth quartiles (Q4) of monthly PM_2.5_, PM_10_ and NO_2_ concentrations from the fetal period to the first 3 years of life compared with the first quartile (Q1) as the reference. Zero in the *x*-axis scale indicates the time of birth, minus numbers indicate the weeks prior to birth and plus numbers indicate the postnatal months. All models were adjusted for monthly mean ambient temperature, household air pollution conditions, child sex and age, maternal and paternal ages at birth, maternal and paternal education, family income, marital status, parity, multivitamin supplementation during pregnancy, gestational diseases, passive smoking during pregnancy, gestation alcohol consumption, feeding pattern, average daily sleep duration for children and frequencies of parent–child interactive activities at 0–1 and 1–3 years old.

**Figure 3 ijerph-19-10482-f003:**
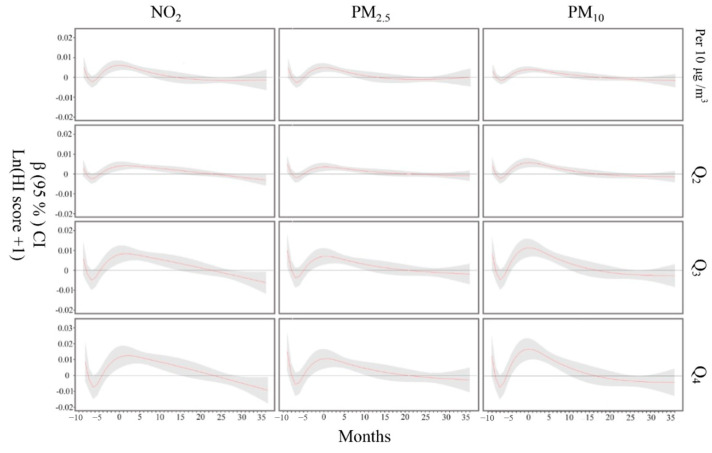
Changes in child ln (HI score + 1) at around 3 years old associated with monthly PM_2.5_, PM_10_ and NO_2_ exposure from fetal period to the first 3 years of life. Note: β (95% CI) in the three uppermost charts indicate the risks of PM_2.5_, PM_10_ and NO_2_ for each 10 μg/m^3^ increment in monthly PM_2.5_, PM_10_ and NO_2_ concentrations during pregnancy. β (95% CI) in the other charts indicate the risks of PM_2.5_, PM_10_ and NO_2_ for the second (Q2), third (Q3) and fourth quartiles (Q4) of monthly PM_2.5_, PM_10_ and NO_2_ concentrations from fetal period to the first 3 years of life compared with the first quartile (Q1) as the reference. Zero in the *x*-axis scale indicates the time of birth, minus numbers indicate the weeks prior to birth and plus numbers indicate the postnatal months. All models were adjusted for monthly mean ambient temperature, household air pollution conditions, child sex and age, maternal and paternal ages at birth, maternal and paternal education, family income, marital status, parity, multivitamin supplementation during pregnancy, gestational diseases, passive smoking during pregnancy, gestation alcohol consumption, feeding pattern, average daily sleep duration for children and frequencies of parent–child interactive activities at 0–1 and 1–3 years old.

**Figure 4 ijerph-19-10482-f004:**
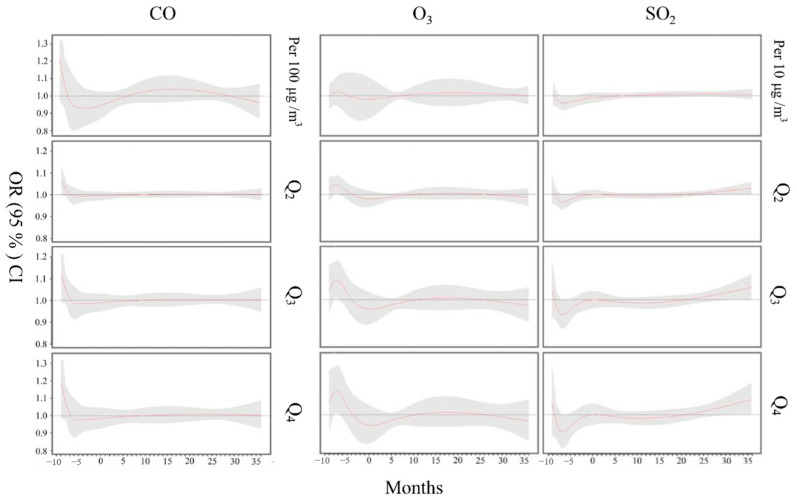
Odds ratio of child hyperactivity with monthly CO, O_3_ and SO_2_ exposure from fetal period to the first 3 years of life. Note: ORs (95% CI) in the three uppermost charts indicate the risks of CO, O_3_ and SO_2_ for each 10 μg/m^3^ increment in monthly CO, O_3_ and SO_2_ concentrations during pregnancy. ORs (95% CI) in the other charts indicate the risks of CO, O_3_ and SO_2_ for the second (Q2), third (Q3) and fourth quartiles (Q4) of monthly PM_2.5_, PM_10_ and NO_2_ concentrations from fetal period to the first 3 years of life compared with the first quartile (Q1) as the reference. Zero in the *x*-axis scale indicates the time of birth, minus numbers indicate the weeks prior to birth and plus numbers indicate the postnatal months. All models were adjusted for monthly mean ambient temperature, household air pollution conditions, child sex and age, maternal and paternal ages at birth, maternal and paternal education, family income, marital status, parity, multivitamin supplementation during pregnancy, gestational diseases, passive smoking during pregnancy, gestation alcohol consumption, feeding pattern, average daily sleep duration for children and frequencies of parent–child interactive activities at 0–1 and 1–3 years old.

**Figure 5 ijerph-19-10482-f005:**
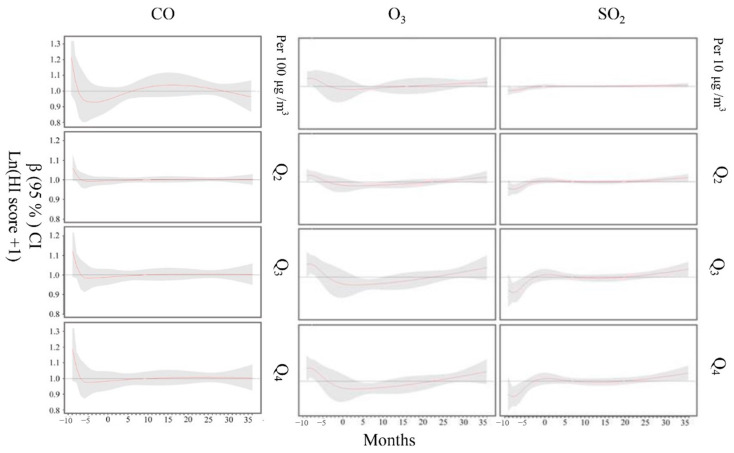
Changes in child ln (HI score +1) around 3 years old associated with monthly CO, O_3_ and SO_2_ exposure from fetal period to the first 3 years of life. Note: β (95% CI) in the three uppermost charts indicate the risks of CO, O_3_ and SO_2_ for each 10 μg/m^3^ increment in monthly CO, O_3_ and SO_2_ concentrations during pregnancy. β (95% CI) in the other charts indicate the risks of CO, O_3_ and SO_2_ for the second (Q2), third (Q3) and fourth quartiles (Q4) of monthly PM_2.5_, PM_10_ and NO_2_ concentrations from fetal period to the first 3 years of life compared with the first quartile (Q1) as the reference. Zero in the *x*-axis scale indicates the time of birth, minus numbers indicate the weeks prior to birth and plus numbers indicate the postnatal months. All models were adjusted for monthly mean ambient temperature, household air pollution conditions, child sex and age, maternal and paternal ages at birth, maternal and paternal education, family income, marital status, parity, multivitamins supplementation during pregnancy, gestational diseases, passive smoking during pregnancy, gestation alcohol consumption, feeding pattern, average daily sleep duration for children and frequencies of parent–child interactive activities at 0–1 and 1–3 years old.

**Table 1 ijerph-19-10482-t001:** Descriptive statistics of parents and child in the analysis (N = 26,052).

Variables	^†^ Mean ± SD or *n* (%)	Variables	Mean ± SD or *n* (%)
Maternal age at child’s birth	28.013 ± 3.973	Gestation drinking	
Paternal age at child’s birth	30.421 ± 4.678	No	25,896 (99.4)
Maternal education		Yes	156 (0.6)
Primary or secondary	3804 (14.6)	Gestational hypertension	
High school	14,250 (54.7)	No	25,583 (98.2)
College and above	7998 (30.7)	Yes	469 (1.8)
Paternal education		Gestational diabetes mellitus	
Primary or secondary	3022 (11.6)	No	24,619 (94.5)
High school	12,244 (47.0)	Yes	1433 (5.5)
College and above	10,786 (41.4)	Preeclampsia/eclampsia	
Maternal occupational		No	25,948 (99.6)
Housewife	3673 (14.1)	Yes	104 (0.04)
Employed	22,379 (85.9)	Taken folic acid during pregnancy	
Family income, RMB/month		No	2293 (8.8)
<5000	2319 (8.9)	Yes	23,759 (91.2)
5001–10,000	5497 (21.1)	Taken calcium during pregnancy	
10,001–20,000	9170 (35.2)	No	3595 (13.8)
>20,000	9066 (34.8)	Yes	22,457 (86.2)
Maternal marital status		Taken vitamins during pregnancy	
Married	25,609 (98.3)	No	15,397 (59.1)
Unmarried/divorced/windowed	443 (1.7)	Yes	10,655 (40.9)
Singleton pregnancy		Feeding pattern	
No	651 (2.5)	Breast feeding	6721 (25.8)
Yes	25,401 (97.5)	Mixed feeding	16,621 (63.8)
Delivery mode		Artificial feeding	2710 (10.4)
Eutocia	14,459 (55.5)	Number of person residence	
Caesarean	11,593 (44.5)	≤4	17,325 (66.5)
Child age	3.506 ± 0.290	>4	8727 (33.5)
Child sex		Single child	
Boy	14,146 (54.3)	No	12,192 (46.8)
Girl	11,906 (45.7)	Yes	13,860 (53.2)

^†^ Mean ± SD or *n* (%) are presented; abbreviations: SD = standard deviation.

**Table 2 ijerph-19-10482-t002:** Summary statistics of monthly exposure to air pollutants and ambient temperature in the total subjects during the entire study period.

Air Pollutants	Mean	SD	Min	P_25_	P_50_	P_75_	Max
CO (mg/m^3^)	0.842	0.150	0.402	0.707	0.830	0.978	1.456
PM_2.5_ (μg/m^3^)	36.161	10.276	11.439	28.248	34.972	43.040	92.666
PM_10_ (μg/m^3^)	54.566	15.501	18.863	43.183	52.967	64.726	113.970
NO_2_ (μg/m^3^)	34.627	9.845	12.477	27.384	33.602	41.070	71.998
O_3_ (μg/m^3^)	93.951	25.108	26.930	74.896	91.375	109.187	197.713
SO_2_ (μg/m^3^)	8.983	2.600	1.285	6.828	8.489	10.471	25.013
Mean temperature (°C)	22.925	5.277	11.247	18.439	23.697	27.934	29.561

**Table 3 ijerph-19-10482-t003:** Cumulative effects of early life exposure to ambient air pollution and child hyperactivity at around 3 years old.

Air Pollutants (10 ug/m^3^)	Total Cumulative Effects		Cumulative Effects during Sensitive Period	
	OR (95% CI)	β (95% CI)	OR (95% CI)	β (95% CI)
PM_10_	1.315 (1.145~1.511)	0.030 (0.020~0.040)	1.296 (1.114~1.508)	0.031 (0.021~0.039)
PM_2.5_	1.335 (1.188~1.500)	0.034 (0.026~0.043)	1.447 (1.175~1.781)	0.033 (0.019~0.048)
NO_2_	1.385 (1.126~1.703)	0.038 (0.023~0.052)	1.387 (1.095~1.757)	0.039 (0.023~0.056)
^†^ CO	1.046 (0.975~1.122)	0.006 (−0.001~0.011)	-	-
SO_2_	0.863 (0.659~1.131)	0.023 (−0.003~0.043)	-	-
O_3_	0.985 (0.946~1.026)	0.010 (−0.003~0.003)	-	-

^†^: CO: 100 ug/m^3^.

## Data Availability

The data presented in this study are available on request from the corresponding author. The data are not publicly available due to being personal health information.

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
