# Peer review of "Fetal Exposure to Air Pollution in Late Pregnancy Significantly Increases ADHD-Risk Behavior in Early Childhood"

_ijerph, 2022, doi:10.3390/ijerph191710482_

Round 1
Reviewer 1 Report
Generally, the manuscript displayed very good scientific features but for a few choices of words or language ambiguities. For example, as follows:
Introduction: Fetal or Fetus Exposure…?
Abstract: In the second line. “…groups of fetal, infants and toddlers.”
The implications of the above for the manuscript is for the authors to know when to use the ADJECTIVE and the NOUN throughout the paper.
Also in the Abstract, can the authors use ‘In particular’ to introduce the second sentence and ‘Moreover’ to start the third sentence? I think these two changes may help further to easily grasp the flow and focus of the manuscript from the onset.
Conclusions: “Conclusion” or Conclusions” section?
Author Response
1) The manuscript had a few choices of words or language ambiguities.
Response: Thank you for your sincere comments. We have modified the words and language ambiguities of this manuscript according to your comments.
2) Can the authors use ‘In particular’ to introduce the second sentence and ‘Moreover’ to start the third sentence?
Response: Thank you for your kind advice. We have added a few conjunctions in the manuscript to make sure our readers easily grasp the flow and focus from the onset.
3) “Conclusion” or “Conclusions” section?
Response: Thank you for your comments. We have revised “Conclusion” to “Conclusions”.

Reviewer 2 Report
This study analyzed the correlation between air pollution concentration and the risk of child ADHD based on a large dataset of 26052 children, which is very impressive. It reported that the exposure udner certain air pollutants (i.e., PM10, PM2.5, NO2) during daily life has positive association with child ADHD-like behavior while other air pollutants such as CO, O3 and SO2 do not have such association. In addition, it has several highlights. For example, it used a deep learning multi-output LSTM model to interpret the air pollution in the year of 2011 to 2012. It also comprehensively reviewed other relevant study and discussed the limitations of present study. I only have a few minor comments:
1) What are the National Environmental Quality Standards for PM10, PM2.5, NO2? Were the levels of PM10, PM2.5, NO2 in the present study higher than that?
2) Does the Table# in the main text refer to the Table S# in the supplementary?
3) What is the map in Fig. S3? Is it a Longhua district map?
Author Response
1)Response: Thank you for your kind comments. According to the Chinese Ambient Air Quality Standard (GB3095-2012) published by the ministry of ecological environment of China(China M E P, 2012. Ambient Air Quality Standards. GB 3095-2012. China Environmental Science Press, Beijing), the National Environmental Quality Standards for PM10, PM2.5, NO2 are 40 ug/m3, 15 ug/m3, 40 ug/m3. The levels of PM10, PM2.5, and NO2 in our study are higher than National Environmental Quality Standards.
2)Response: Thank you for your kind comments. The Table# in the main text refers to the tables in the “Tables and figures” file. We have uploaded it with the manuscript together.
3)Response: Thank you for your sincere comments. Fig.S3 is not Longhua district maps, but prediction maps of the six air pollutant concentrations of Shenzhen generated using regression kriging. Longhua District covers only 175.6 square kilometers and has a big adjustment in its administrative division in 2017. Therefore, we choose the air pollution prediction maps of Shenzhen to fully display air pollution differences and ensure consistency of map boundaries. Meanwhile, considering our study contains 504 maps (6 air pollutant×12months×7years), we choose March 2013 as an example.
